# Effect of changing $NO_x$ lifetime on the seasonality and long-term trends of satellite-observed tropospheric $NO_2$ columns over China

Viral Shah[1], Daniel J. Jacob[1,2], Ke Li[1], Rachel F. Silvern[2], Shixian Zhai[1], Mengyao Liu[3], Jintai Lin[3], and Qiang Zhang[4]

[1]Harvard John A. Paulson School of Engineering and Applied Sciences, Harvard University, Cambridge, MA, USA.
[2]Department of Earth and Planetary Sciences, Harvard University, Cambridge, MA, USA.
[3]Laboratory for Climate and Ocean-Atmosphere Studies, Department of Atmospheric & Oceanic Sciences, School of Physics, Peking University, Beijing, China.
[4]Department of Earth System Science, Tsinghua University, Beijing, China.

*Correspondence to*: Viral Shah (vshah@seas.harvard.edu)

**Abstract.** Satellite observations of tropospheric $NO_2$ columns are extensively used to infer trends in anthropogenic emissions of nitrogen oxides ($NO_x \equiv NO + NO_2$), but this may be complicated by trends in $NO_x$ lifetime. Here we use 2004–2018 observations from the OMI satellite-based instrument (QA4ECV and POMINO v2 retrievals) to examine the seasonality and trends of tropospheric $NO_2$ columns over central-eastern China, and we interpret the results with the GEOS-Chem chemical transport model. The observations show a factor of 3 increase in $NO_2$ columns from summer to winter, which we explain in GEOS-Chem as reflecting a longer $NO_x$ lifetime in winter than in summer (21 h versus 5.9 h in 2017). The 2005–2018 summer trends of OMI $NO_2$ closely follow the trends in the Multi-resolution Emission Inventory for China (MEIC), with a rise over the 2005–2011 period and a 25% decrease since. We find in GEOS-Chem no significant trend of the $NO_x$ lifetime in summer, supporting the emission trend reported by MEIC. The winter trend of OMI $NO_2$ is steeper than in summer over the entire period, which we attribute to a decrease in $NO_x$ lifetime at lower $NO_x$ emissions. Half of the $NO_x$ sink in winter is from $N_2O_5$ hydrolysis, which counterintuitively becomes more efficient as $NO_x$ emissions decrease due to less titration of ozone at night. Formation of organic nitrates also becomes an increasing sink of $NO_x$ as $NO_x$ emissions decrease but emissions of volatile organic compounds (VOCs) do not.

## 1 Introduction

Emissions of nitrogen oxides ($NO_x \equiv NO+NO_2$) from fossil fuel combustion in China have been changing fast in the past few decades, due to rapid economic expansion on the one hand and strengthening environmental regulations on the other hand. Growing fossil fuel use with weak pollution controls resulted in almost three-fold increase in China's $NO_x$ emissions between 1990 and 2010, according to bottom-up inventories based on activity data and emission factors (Granier et al., 2011; Zhang et al., 2012b). Since then, China has adopted strong measures to decrease air pollution by setting stringent emissions standards,

capping coal use, increasing vehicle fuel efficiency, closing outdated facilities, and growing renewable energy (Liu et al., 2016; Zheng et al., 2018). Bottom-up inventories estimate that China's $NO_x$ emissions decreased by 20% between 2011 and 2017, despite continuing economic expansion (Sun et al., 2018; Zheng et al., 2018). There is a strong need to evaluate these emission inventories and their trends for air quality management.

Satellite-based observations of tropospheric $NO_2$ columns by solar backscatter have been used extensively as a proxy for $NO_x$ emissions and their trends (Martin, 2008; Streets et al., 2013). These observations have been qualitatively consistent with the trends in Chinese $NO_x$ emission inventories, showing an increasing trend of $NO_2$ columns over China between 1994 and 2011, and a sharp reversal in eastern China since 2011 (Richter et al., 2005; van der A et al., 2006, 2008; Lin et al., 2010; Krotkov et al., 2016; Schneider et al., 2015; Duncan et al., 2016; Cui et al., 2016; Georgoulias et al., 2019; Wang et al., 2019). However, the trends in the $NO_2$ columns are steeper than in the emission inventories (Zhang et al., 2007; Stavrakou et al., 2008; Zhang et al., 2012b; Hilboll et al., 2013; Liu et al., 2017; Zheng et al., 2018). For example, Zhang et al. (2007) found that $NO_2$ columns over China increased two-fold from 1996 to 2004 while their emission inventory reported a 60% increase. For 2010–2015, the Multi-resolution Emission Inventory for China (MEIC) estimates a 14% decrease in $NO_x$ emissions but $NO_2$ columns from the OMI satellite instrument indicate a 22% decrease (Zheng et al., 2018).

Differences in the trends between $NO_2$ columns and $NO_x$ emission inventories could reflect errors in the inventories (Saikawa et al., 2017) and satellite retrievals (Lin et al., 2014; Lorente et al., 2017), but also trends in the lifetime of $NO_x$ against atmospheric oxidation. This lifetime is of the order of hours and may change with the chemical environment, including the $NO_x$ concentration itself (Stavrakou et al., 2008; Lamsal et al., 2011; Valin et al., 2011; Lu and Streets, 2012; Duncan et al., 2013; Gu et al., 2016; Cooper et al., 2017; Laughner and Cohen, 2019). $NO_x$ is oxidized to nitric acid ($HNO_3$) and organic nitrates ($RONO_2$), including peroxyacyl nitrates ($RC(O)OONO_2$). There is also a minor sink from $NO_2$ dry deposition (Zhang et al., 2012a). Oxidation in the daytime is driven by photochemically produced hydrogen oxide ($HO_x \equiv OH + HO_2 + RO_2$) radicals:

$$NO_2 + OH \xrightarrow{M} HNO_3 \qquad (R1)$$

$$NO + RO_2 \xrightarrow{\alpha\,M} (1-\alpha)NO_2 + (1-\alpha)RO + \alpha RONO_2 \qquad (R2)$$

$$NO_2 + RC(O)OO \xleftrightarrow{M} RC(O)OONO_2 \qquad (R3)$$

where $\alpha$ in reaction R2 is the branching ratio for organic nitrate formation. $NO_x$ and $HO_x$ concentrations are tightly interlinked (Kleinman, 1994; Laughner and Cohen, 2019). When $NO_x$ levels are sufficiently low (the so-called $NO_x$-limited conditions), an increase in $NO_x$ drives an increase in $HO_x$, particularly OH through the $NO + HO_2 \rightarrow NO_2 + OH$ reaction. When $NO_x$ levels

are high (the so-called $NO_x$-saturated conditions), reaction R1 becomes the dominant $HO_x$ sink and an increase in $NO_x$ causes a decrease in $HO_x$.

At night, the chemical loss of $NO_x$ proceeds through a series of reactions (R4–R10) beginning with the oxidation of NO and $NO_2$ by $O_3$ to form nitrate radical ($NO_3$) and dinitrogen pentoxide ($N_2O_5$). $N_2O_5$ and $NO_3$ react in aerosols to produce $HNO_3$, and $NO_3$ additionally reacts with volatile organic compounds (VOCs) to form either $HNO_3$ or $RONO_2$:

$$NO + O_3 \longrightarrow NO_2 + O_2 \tag{R4}$$

$$NO_2 + O_3 \longrightarrow NO_3 + O_2 \tag{R5}$$


$$NO_3 + NO_2 \overset{M}{\longleftrightarrow} N_2O_5 \tag{R6}$$

$$N_2O_5 \xrightarrow{H_2O(l),\ aerosols} 2HNO_3 \tag{R7}$$

$$N_2O_5 + Cl^- \xrightarrow{H_2O(l),\ aerosols} HNO_3 + ClNO_2 \tag{R8}$$

$$NO_3 \xrightarrow{H_2O(l),\ aerosols} HNO_3 + OH \tag{R9}$$

$$NO_3 + VOC \longrightarrow HNO_3, RONO_2, \text{other products} \tag{R10}$$

A change in $NO_x$ emissions can change the nighttime levels of ozone and reaction (R6) is quadratic in $NO_x$ concentrations, leading to nonlinearity between $NO_x$ emissions and $NO_2$ concentrations. In addition, aerosol concentrations in China have decreased by about 30% since 2013 (Lin et al., 2018; Zheng et al., 2018; Zhai et al., 2019; Ma et al., 2019), which would decrease the rate of nighttime $NO_x$ loss (R7–R9). A decrease in aerosol concentrations will also slow the $NO_x$ loss by hydrolysis of $NO_2$ in aerosols:


$$NO_2 \xrightarrow{H_2O(l),\ aerosols} 0.5\ HONO + 0.5\ HNO_3 \tag{R11}$$

Here we present trends in tropospheric $NO_2$ columns over China from 2004 to 2018 observed by the Ozone Monitoring Instrument (OMI) satellite instrument, showing a peak in 2011. We use the GEOS-Chem chemical transport model applied to the MEIC inventory (Zheng et al., 2018) to investigate how changes in $NO_x$ lifetimes have affected the relationship between tropospheric $NO_2$ columns and $NO_x$ emissions, and whether this can reconcile the differences in trends between the two

quantities. Results have important implications for the use of satellite $NO_2$ retrievals to infer trends in $NO_x$ emissions.

## 2 Observations and model

### 2.1 OMI NO$_2$ column retrievals

We use 2004–2018 tropospheric NO$_2$ column data retrieved from the OMI instrument aboard the NASA Aura satellite. Aura is in sun-synchronous polar orbit satellite with daytime equator crossing at 13:45 local solar time (LST). OMI measures backscattered solar radiation from the Earth in the ultraviolet and visible wavelength range (270–504 nm). It has a swath width of 2600 km and a ground pixel size of 13 km × 24 km at nadir (Levelt et al., 2006, 2018). Several OMI pixels are affected by the so-called row anomaly possibly from an obstruction in their field of view (Dobber et al., 2008). Pixels not affected by the row anomaly have no significant calibration drift over the length of the record (Boersma et al., 2018).

We use tropospheric NO$_2$ columns from two retrievals: the European Quality Assurance for Essential Climate Variables (QA4ECV) project's NO$_2$ ECV precursor product (Boersma et al., 2018) and the Peking University POMINO product, version 2 (Lin et al., 2015; Liu et al., 2019). NO$_2$ tropospheric column retrieval in the ECV product involves (1) spectral fit of the backscattered solar radiation in the 405–465 nm window to obtain the total NO$_2$ slant column (SC), (2) removal of the stratospheric component by data assimilation with the TM5-MP chemical transport model, and (3) conversion of the tropospheric SC to a tropospheric vertical column (VC) with an air mass factor (AMF = SC/VC) that depends on viewing geometry, surface albedo, retrieved cloud properties, and the NO$_2$ vertical profile (taken from the TM5-MP model).

The POMINO v2 product starts with the tropospheric NO$_2$ slant columns from the ECV retrieval but uses different methods and data sources for the AMF calculation. The main difference is the treatment of aerosols. ECV assumes that aerosol effects are implicitly accounted for in the independent retrieval of cloud pressure and cloud fraction, which are prerequisites for NO$_2$ retrievals. POMINO explicitly accounts for aerosols in the radiative transfer calculations with aerosol optical properties and vertical profiles from the GEOS-Chem model corrected with satellite observations from the MODIS and CALIOP instruments. In polluted areas (aerosol optical depth greater than 0.5), the choice of aerosol correction method can affect the AMF by 45% (Lorente et al., 2017) and the sign of the correction depends on the vertical distribution of aerosols relative to NO$_2$ (Palmer et al., 2001; Martin et al., 2003; Liu et al., 2019). POMINO also includes angular dependence of surface reflectance, online radiative transfer calculation, and consistency in retrievals of cloud properties and NO$_2$ (Lin et al., 2015; Liu et al., 2019).

We create monthly mean gridded (0.5º latitude × 0.625º longitude) datasets of ECV and POMINO NO$_2$ columns over China for June-July-August (JJA) 2005–2018 and December-January-February (DJF) 2004–2018. We exclude pixels with snow-covered surfaces and cloud fraction greater than 30%. We use cloud fraction data from the corresponding retrievals. We include only cross-track viewing positions of 5 through 22 to exclude data affected by the row anomaly (Boersma et al., 2018) and swath edges. For comparison with GEOS-Chem, the POMINO and ECV NO$_2$ columns are recalculated with modified air mass factors using the pixel-specific GEOS-Chem NO$_2$ vertical profiles (Palmer et al., 2001).

## 2.2 Ground-based observations

We use hourly measurements of $NO_2$ and $O_3$ concentrations from the network of ~1000 sites operated by the China Ministry of Ecology and Environment (MEE) since 2013 (http://beijingair.sinaapp.com). We correct the known interference of organic nitrates and $HNO_3$ in the $NO_2$ measurements by using the GEOS-Chem simulated concentrations for those species following

Lamsal et al. (2008) and then grid (0.5º×0.625º grid) and seasonally average the data, discarding sites with less than 50% coverage in a season. The correction for $HNO_3$ and organic nitrates decreases the reported seasonal-mean $NO_2$ concentrations over eastern China by 25% in summer and 6% in winter.

## 2.3 GEOS-Chem model

We use the GEOS-Chem chemical transport model version 12.1.0 (www.geos-chem.org) driven by assimilated meteorological

fields from the NASA Global Modeling and Assimilation Office's Modern-Era Retrospective analysis for Research and Applications, version 2 (MERRA-2) system (Gelaro et al., 2017). GEOS-Chem simulates the chemistry of major gas- and aerosol-phase species in the troposphere (Pye et al., 2009; Kim et al., 2015; Travis et al., 2016; Fisher et al., 2016; Sherwen et al., 2016). We use the GEOS-Chem Classic nested-grid configuration over East Asia (11ºS–55ºN, 60–150ºE) at 0.5º×0.625º resolution (Wang et al., 2004; Chen et al., 2009), with lateral chemical boundary conditions from a 4º×5º global simulation.

Anthropogenic emissions over China are from the MEIC inventory updated annually for 2000–2017 (www.meicmodel.org; Zheng et al., 2018). MEIC includes monthly emission profiles for each sector (Li et al., 2017a) and hourly profiles developed at Tsinghua University. We vertically resolve emissions from point sources (power plants and industries) following profiles used in the LOTOS-EUROS model (Manders et al., 2017), and speciate anthropogenic $NO_x$ emissions as NO (90%), $NO_2$ (9.2%) and HONO (0.8%) following Menut et al. (2013). GEOS-Chem includes additional $NO_x$ emissions from soil and

fertilizer use (Hudman et al., 2012), lightning (Murray et al., 2012), shipping (Vinken et al., 2011; Holmes et al., 2014) and aircraft (Stettler et al., 2011). Vertical mixing in the planetary boundary layer is simulated using a non-local mixing scheme (Lin and McElroy, 2010).

We modified the standard GEOS-Chem version 12.1.0 chemistry to update the reactive uptake coefficients (γ) of $N_2O_5$, $NO_3$, and $NO_2$ on aerosols (Jacob, 2000) based on recent comparison of GEOS-Chem to observations from the Wintertime

Investigation of Transport, Emissions, and Reactivity (WINTER) aircraft campaign over the eastern United States (Jaeglé et al., 2018; Shah et al., 2018). $\gamma_{N_2O_5}$ is computed following Bertram and Thornton (2009) for sulfate-nitrate-ammonium aerosols, and is taken to be $1\times10^{-4}$ (RH < 50%) or $1\times10^{-3}$ (RH > 50%) for organic aerosols. $N_2O_5$ hydrolysis produces $HNO_3$ and $ClNO_2$ on sea-salt aerosols with a 1:1 branching ratio (reaction R8) and only $HNO_3$ on other aerosol types (reaction R7). Uniform values of $\gamma_{N_2O_5}$ and $\gamma_{NO_2}$ are used for all aerosol types and all RH conditions. $\gamma_{NO_3}$ is taken to be $1\times10^{-3}$ following Jacob (2000).

$\gamma_{NO_2}$ for the hydrolysis reaction (R11) is decreased from $1\times10^{-4}$ (Jacob, 2000) to $1\times10^{-5}$ on the basis of laboratory measurements

(Bröske et al., 2003; Stemmler et al., 2007; Tan et al., 2016). This decrease of $\gamma_{NO_2}$ yields a 24 h mean wintertime $HONO/NO_2$ molar concentration ratio of 0.035 over eastern China in GEOS-Chem, consistent with the observed range of 0.015–0.071 (Hendrick et al., 2014; Spataro et al., 2013; Wang et al., 2017, 2013). The GEOS-Chem $HONO/NO_2$ ratio is likely low because of unknown sources of HONO particularly during the day (Kleffmann, 2007; Spataro and Ianniello, 2014). For the WINTER campaign, Jaeglé et al. (2018) used a $\gamma_{NO_2}$ of $1\times10^{-4}$ but assumed that the reaction produces only HONO. Using this $\gamma_{NO_2}$ for eastern China would lead to a significant overestimate of the observed $HONO/NO_2$ ratio.

We evaluate the model with the spatial and seasonal distributions of OMI $NO_2$ observations for 2016/17 DJF and 2017 JJA (the latest year with MEIC data) and use these two periods to analyze the seasonality of $NO_x$ lifetime and loss pathways in the model. To calculate the emission-driven changes in $NO_x$ lifetimes, we conduct a sensitivity simulation in which we set anthropogenic emissions over China to 2012 levels but use the 2016/17 meteorology and $NO_x$ emissions from soils, lightning, ships, and aircraft. Chinese $NO_x$ emissions decreased by 25% from 2012 to 2017 according to MEIC (Supplement Fig. S1). For comparison with OMI observations, we sample the model at 13–14 LST and exclude model columns with surface snow cover or with model cloud fraction greater than 30%. We focus on the large polluted region of central-eastern China (30–41ºN, 112–122ºE; rectangles in Fig. 1), where we may expect tropospheric $NO_2$ columns to be most sensitive to Chinese $NO_x$ emissions, and where the relatively narrow latitude range leads to consistent seasonal variations. This region accounted for 50% of anthropogenic Chinese emissions in 2017 according to MEIC.

## 3 Results and discussion

### 3.1 Seasonal variation of $NO_2$ columns and $NO_x$ lifetimes

Figure 1 shows the $NO_2$ columns from the POMINO and ECV retrievals, and from the GEOS-Chem model, for JJA 2017 and DJF 2016/17. In central-eastern China, we find that in both seasons over 70% of the GEOS-Chem tropospheric $NO_2$ column as would be observed by OMI is in the boundary layer below 2 km altitude. Thus we expect the $NO_2$ column to reflect mostly the local $NO_x$ emissions rather than the free tropospheric background (Silvern et al., 2019). In summer, average GEOS-Chem $NO_2$ columns over central-eastern China are within 10% of the POMINO and ECV $NO_2$ columns. There is scatter in the spatial relationship ($r \approx 0.5$) that could be due to a combination of model and retrieval errors. In winter, however, POMINO is 42% higher than GEOS-Chem while ECV is 16% lower. The difference in aerosol correction between POMINO and ECV is largest in winter, due to high aerosol concentrations and high solar zenith angles. In ECV, polluted scenes with high aerosol optical depths (and likely high $NO_2$) can be misclassified as cloudy and excluded from the seasonal-mean, which leads to a negative sampling bias (Lin et al., 2014, 2015; Liu et al., 2019). On the other hand, retrieving $NO_2$ columns under high-aerosol conditions can be uncertain because of the strong sensitivity to the vertical distribution of $NO_2$ relative to that of aerosols,

although this is less of a problem in POMINO as it uses the CALIOP-observed aerosol vertical profiles (Lin et al., 2014, 2015; Liu et al., 2019).

Liu et al. (2019) compared the ECV and POMINO retrievals to ground-based Multi Axis Differential Optical Absorption Spectroscopy (MAX-DOAS) $NO_2$ column observations on 49 days in 3 Chinese cities. POMINO was on average closer to the MAX-DOAS $NO_2$ than ECV (-3% versus -22% bias) and on hazy days (+4% versus -26% bias), but on clear (cloud fraction=0)

days ECV performed better (-6% bias) than POMINO (+30% bias). These biases were slightly larger in fall and winter, although sampling in individual seasons was sparse. There is uncertainty in the comparison as the column observed by MAX-DOAS may not be representative of the satellite pixel, and aerosol vertical profiles used in the satellite and MAX-DOAS retrievals may be inconsistent (Lin et al., 2014).

Figure 2 compares the mean winter-summer ratios of $NO_2$ columns from the ECV and POMINO retrievals over central-eastern

China with GEOS-Chem and with the ratios of 24-h mean surface $NO_2$ concentrations at the MEE sites. GEOS-Chem shows similar ratios for the afternoon $NO_2$ columns and 24-h mean surface $NO_2$, despite different averaging times. We find that the seasonal amplitude in the surface $NO_2$ data is most consistent with the ECV $NO_2$ columns, whereas the seasonal amplitude of POMINO $NO_2$ is larger. Winter-summer ratios of $NO_2$ columns are 2.6 in the ECV retrieval, 3.5 in POMINO, and 2.7 in GEOS-Chem. Winter-summer ratios of surface $NO_2$ concentrations are 2.3 in the MEE data and 2.5 in GEOS-Chem.

Anthropogenic $NO_x$ emissions in the MEIC inventory show little seasonality, with a winter-summer ratio is only 1.15 (Li et al., 2017b). We find in GEOS-Chem that the seasonal variation in $NO_2$ columns is mainly driven by the $NO_x$ lifetime in the boundary layer, which increases from 5.9 h in summer to 21 h in winter (24-h mean lifetimes for 2017, Fig. 3). We find little difference between the boundary layer and the tropospheric column $NO_x$ lifetimes (Supplement Fig. S2) since most of the column is in the boundary layer.

In summer, $NO_x$ is lost mostly through oxidation by OH in daytime (43%) and through $N_2O_5$ hydrolysis forming $HNO_3$ at night (33%). The daytime oxidation processes have a larger effect on the afternoon $NO_2$ columns because of the short $NO_x$ lifetime, particularly in summer. In winter, the $NO_x$ lifetime is much longer because of the lower concentrations of OH and $RO_2$ radicals. $N_2O_5$ hydrolysis accounts for 51% of $NO_x$ loss in winter. Remarkably, the loss of $NO_x$ from $N_2O_5$ hydrolysis is a factor of 2 slower in winter than in summer, despite the longer nights and higher aerosol concentrations (Zhai et al., 2019),

because of the low nighttime ozone concentrations. The overall $NO_x$ lifetime in winter and the contribution from $N_2O_5$ hydrolysis are similar to values inferred over the eastern US during the WINTER campaign (Jaeglé et al., 2018), despite a factor of ~5 difference in aerosol concentrations between the two regions. Loss of $NO_x$ by $N_2O_5$ hydrolysis in China in winter is limited by the supply of ozone, not the supply of aerosol.

The modeled $NO_x$ lifetimes can be affected by uncertainties in the modeled aerosol surface area, $\gamma_{N_2O_5}$, and the yield of $ClNO_2$.
We find that the GEOS-Chem aerosol concentrations at the surface are about 30% higher than observations from the MEE network. On the other hand, GEOS-Chem's $\gamma_{N_2O_5}$ values are 25% lower on average than the observation-based estimates from the WINTER campaign (Jaeglé et al., 2018; McDuffie et al., 2018b). We tested the sensitivity of the modeled $NO_x$ lifetime to the aerosol surface area and $\gamma_{N_2O_5}$ with a separate simulation and find that a 30% change in either quantity changes the $NO_x$ loss by $N_2O_5$ hydrolysis by only 8% in summer and less than 2% in winter. We assume that the yield of $ClNO_2$ from $N_2O_5$ hydrolysis for all aerosols other than sea-salt is 0, which is lower than the average value of ~0.3 estimated by field studies in China (McDuffie et al., 2018a; Tham et al., 2018). If we had assumed a higher $ClNO_2$ yield in the model, the loss of $NO_x$ by $N_2O_5$ hydrolysis would be slower, particularly in winter when it is limited by ozone and the $HNO_3$ and $ClNO_2$ branches compete for the limited $N_2O_5$, and the daytime $NO_x$ loss would increase because of the additional daytime $NO_2$ from $ClNO_2$ photolysis. Sub-grid scale variability in $NO_x$ and oxidant concentrations can also affect the modeled $NO_x$ lifetimes because of the non-linearity of $NO_x$ chemistry. For example, $NO_x$ lifetime within a concentrated $NO_x$ plume at night can be long because of oxidant depletion, and assuming that the plume is instantaneously diluted within the model grid cell can result in a shorter $NO_x$ lifetime (Brown et al., 2012). However, studies that explicitly treat sub-grid scale plumes suggest that this effect can be important locally near large sources but is small at the regional scale (Karamchandani et al., 2011).

Leue et al. (2001) previously estimated the $NO_x$ lifetime in the eastern US by using the offshore gradient of satellite-observed $NO_2$ columns. We tried this approach and found that the offshore gradients of $NO_2$ columns perpendicular to the east coast of China are consistent between model and observations (Supplement Fig. S3). However, there is little seasonal difference in the gradients, suggesting that their magnitudes are determined by dilution more than chemical decay.

### 3.2 2004–2018 trends in NO₂ columns and lifetimes

Figure 4 shows the trends in the summer and winter $NO_2$ columns from the ECV retrieval, and in anthropogenic $NO_x$ emissions from MEIC, over central-eastern China for the 2004–2018 extent of the OMI record. According to MEIC, $NO_x$ emissions increased at 5–6% $a^{-1}$ in 2005–2011 and decreased at the same pace after 2011. OMI $NO_2$ columns mirror the trajectory of the MEIC $NO_x$ emissions: rising between 2005 and 2011, reversing direction in 2011/12, and falling back to around 2005 levels by 2018. Zheng et al. (2018) also showed consistency in the trends of OMI $NO_2$ columns and the MEIC $NO_x$ emissions, but found that the $NO_2$ trends at the MEE surface sites are flatter because the sites are urban and more sensitive to vehicular $NO_x$ emissions. The summer trends in OMI $NO_2$ closely track the MEIC emissions but the winter trends are steeper. The same summer-winter differences in $NO_2$ column trends over China are seen in the POMINO retrieval (Supplement Fig. S4) and in retrievals from the Ozone Mapping Profiler Suite (OMPS) instrument (Lin et al., 2019). Previous studies also reported such a seasonal difference between summer and winter $NO_2$ column trends (Uno et al., 2007; Zhang et al., 2007; Stavrakou et al., 2008; Gu et al., 2013).

The steeper slopes in winter could reflect a trend in $NO_x$ lifetime as $NO_x$ and other emissions change. A few modeling studies have previously explored this dependence for China. Uno et al. (2007) found no significant change in the annual mean $NO_x$ lifetime over 1996–2004. Stavrakou et al. (2008) found that the increase in $NO_x$ emissions over 1997–2006 drove a 25% decrease in summer $NO_x$ lifetime, due to higher OH from faster $NO+HO_2$ reaction, and a 10% increase in winter $NO_x$ lifetime, due to lower OH from faster $NO_2+OH$ reaction.

We examined the effect of 2012–2017 changes in MEIC emissions for $NO_x$ and other species on the lifetime of $NO_x$ simulated by GEOS-Chem (Fig. 3). During that period, $NO_x$ emissions in central-eastern China decreased by 25% and boundary layer aerosol concentrations in GEOS-Chem decreased by 20%. Observed aerosol concentrations from the MEE network decreased by 30% over the 2013–2017 period (Zhai et al., 2019). We find no significant change in the summer $NO_x$ lifetime between 2012 and 2017. The $NO_x$ lifetime during the day shortened slightly, as summertime OH concentrations increased by 6% and
$RO_2$ concentrations increased by 13%. However, the $NO_x$ lifetime during the night increased as aerosol concentrations dropped, canceling the overall effect.

In contrast, the winter $NO_x$ lifetime decreased by 22% (from 27 h to 21 h) between 2012 and 2017 (Fig. 3), driven mostly by faster loss by $N_2O_5$ hydrolysis in aerosols, and also by faster loss from reactions with OH and $RO_2$. The loss rate from $RO_2$ + $NO/NO_2$ is largely determined by the emission rate of VOCs, which changed by less than 3% over 2012–2017 (Zheng et al.,
2018), effectively decreasing the $NO_x$ lifetime as $NO_x$ emissions dropped. At lower $NO_x$ emissions, $NO_x$ loss by $NO_3+VOC$ reactions also increased relative to $NO_x$ loss by $N_2O_5$ hydrolysis, because of an increase in the ratio of $NO_3$ to $N_2O_5$ concentrations. However, the contribution of $NO_3+VOC$ reactions to the total $NO_x$ loss remains less than 5% because of high $NO_2$. The faster loss by $N_2O_5$ hydrolysis in 2017 relative to 2012 might seem counterintuitive since both aerosols and $NO_x$ dropped over the period. However, winter aerosol levels for converting $N_2O_5$ to $HNO_3$ remain in excess even in 2017. Instead,
we find that the driving factor behind the increase in $N_2O_5$ hydrolysis is a rise in nighttime ozone concentrations. At night, fast titration by NO (reaction R4) is an important sink of ozone close to $NO_x$ sources. As $NO_x$ emissions decrease, less ozone is titrated, which then enables the formation of $NO_3$ by reaction (R5) and subsequent formation of $N_2O_5$. The simulated $NO_x$ lifetime against loss by $N_2O_5$ hydrolysis decreases by 26% from 54 h in the winter 2011/12 to 40 h in the winter 2016/17. Ground-based observations at the MEE sites show an increase in winter nighttime ozone consistent with the model (Fig. 5).

Figure 4 shows the implications of these changes in seasonal $NO_x$ lifetime on the 2012–2017 $NO_2$ column trends simulated by GEOS-Chem. Neither the 24-h mean nor the daytime $NO_x$ lifetime changes significantly in summer and therefore the $NO_2$ columns track the MEIC emission trends, consistent with observations. In winter, the shortening of the $NO_x$ lifetime steepens the trends of $NO_2$ columns relative to $NO_x$ emissions. Wintertime GEOS-Chem $NO_2$ columns decrease by 35% between 2011/12 and 2016/17, faster than the 25% decrease in the MEIC $NO_x$ emissions. The $NO_x$ lifetime in winter is about one day,
long enough for faster $NO_x$ loss at night to affect $NO_2$ columns in the afternoon when OMI makes its observations. Comparison

to the observed wintertime trend suggests that the GEOS-Chem decrease in $NO_x$ lifetime over 2012–2017 may not be steep enough. There is substantial uncertainty in the factors controlling wintertime OH concentrations (Tan et al., 2018; Miao et al., 2018) and these might also affect the model trend. Meteorological variability can also cause interannual changes in wintertime $NO_2$ of ~20% (Lin and McElroy, 2011), but the effect on longer-term trends will be smaller. GEOS-Chem trends for 2005–2012 should be symmetric to those for 2012–2017, since $NO_x$ emissions in 2005 are similar to those in 2017 (Fig. 4).

The use of satellite-based $NO_2$ column observations to evaluate trends in $NO_x$ emission inventories in China can be compared to similar work previously done for the US. Jiang et al. (2018) found that OMI $NO_2$ columns over the US during 2009–2015 decreased slower than $NO_x$ emissions in the US Environment Protection Agency's (EPA) National Emissions Inventory (NEI), suggesting that $NO_x$ emission controls were not as effective as reported by the NEI. However, Silvern et al. (2019) explained this discrepancy by a large relative contribution of the free tropospheric background to the $NO_2$ column over the US, weakening the relationship between $NO_2$ columns and US anthropogenic $NO_x$ emissions. This is not a major concern over central-eastern China, where the contribution of the free troposphere above 2 km to the tropospheric $NO_2$ column as sensed by OMI is less than 30%. Laughner & Cohen (2019) find an increase in summer $NO_x$ lifetime over 2010–2013 in OMI observations of isolated urban plumes over the US, reflecting $NO_x$-limited conditions where OH concentrations decrease as $NO_x$ decreases. This would dampen the response of $NO_2$ columns to emission reductions. Such an effect is not apparent in central-eastern China, which is prevailingly in the so-called transition regime between $NO_x$-saturated and $NO_x$-limited conditions (Jin and Holloway, 2015; Li et al., 2019).

## 4 Conclusions

We examined the seasonality and trends of satellite-derived tropospheric $NO_2$ columns over China and their relation to $NO_x$ emissions. Observations from the satellite-based OMI instrument show a factor of 3 increase in tropospheric $NO_2$ columns from summer to winter, and we show that this can be explained by the seasonal variation in $NO_x$ lifetime against oxidation. $NO_2$ columns for the 2004–2018 duration of the OMI record peak in 2011 and subsequently decrease, consistent with the Multi-resolution Emission Inventory for China (MEIC). The summer trends in OMI $NO_2$ columns match closely the MEIC emission trends, but the winter trends are steeper than MEIC. We attribute the steeper winter trends to a decrease in the $NO_x$ lifetime, mostly by faster $N_2O_5$ hydrolysis in aerosols, as $NO_x$ emissions decrease. Lower $NO_x$ emissions lead to an increase in nighttime ozone in winter, promoting $N_2O_5$ formation. Our analysis of the OMI $NO_2$ column observations thus supports the magnitude and trends of $NO_x$ emissions in the MEIC inventory, while emphasizing the need to account for changes in $NO_x$ lifetime when interpreting trends in satellite $NO_2$ columns in terms of trends in $NO_x$ emissions.

**Data availability.** The QA4ECV data is available at http://www.qa4ecv.eu, POMINO v2 at http://www.phy.pku.edu.cn/~acm/acmProduct.php#POMINO, the MEE surface data at http://beijingair.sinaapp.com and the MEIC inventory data at http://www.meicmodel.org. GEOS-Chem results are available on request from the corresponding author.

**Author contributions.** VS and DJJ designed the study. VS performed the model simulations and data analysis. KL and SZ
processed the ground-based observations. RFS contributed analysis software. ML and JL provided the POMINO data. QZ provided the MEIC data. VS and DJJ wrote the paper with contributions from all co-authors.

**Competing interests.** The authors declare that they have no conflict of interest.

**Acknowledgments.** This work was funded by the NASA Earth Science Division. The development of POMINO product was funded by the National Natural Science Foundation of China (41775115). We acknowledge the QA4ECV project for the $NO_2$
data.

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

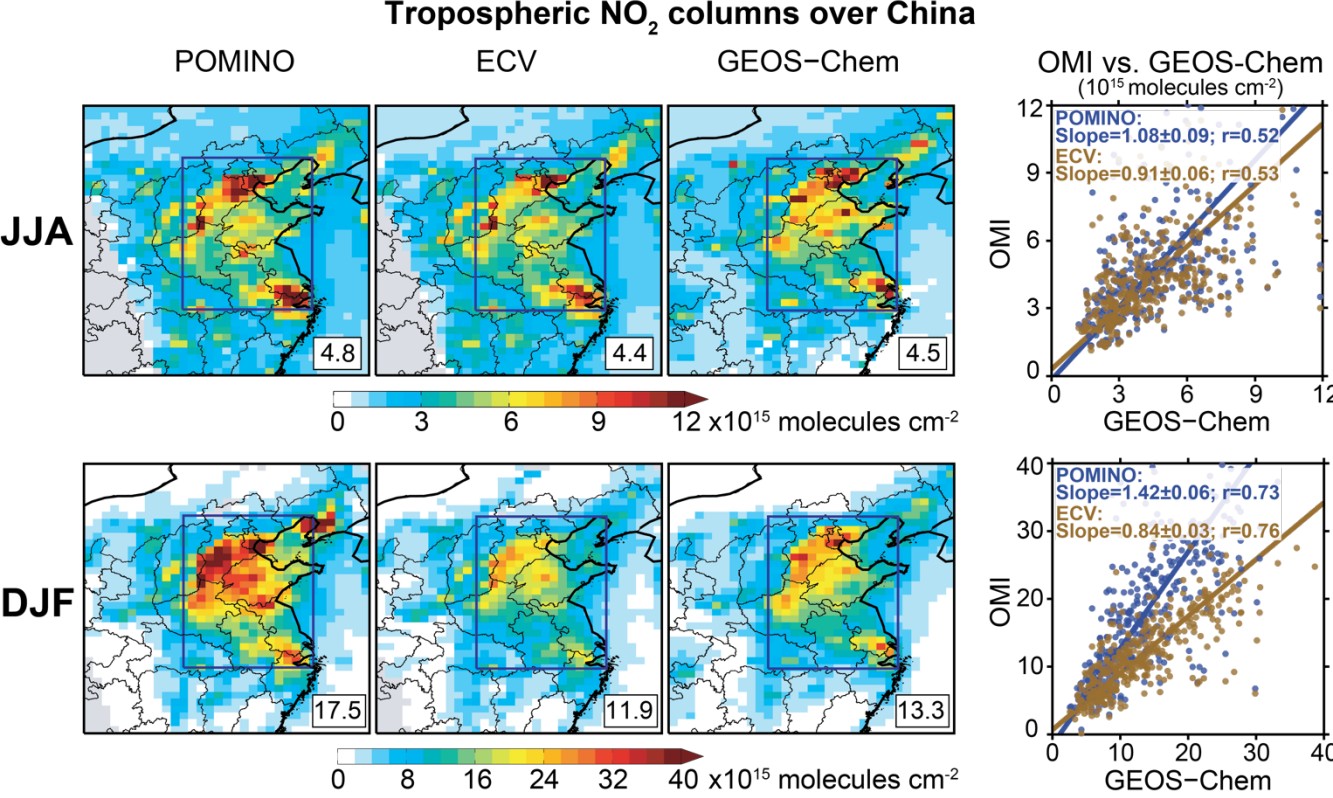

**Figure 1: Tropospheric NO₂ columns over China.** Values are 3-month means for June-July-August (JJA) 2017 and December-January-February (DJF) 2016/17 on the 0.5°×0.625° GEOS-Chem grid. OMI retrievals from POMINO (Liu et al., 2019) and ECV (Boersma et al., 2018) are compared with the GEOS-Chem model. The JJA and DJF panels have different color scales. Locations where none of the OMI data met our selection criteria are in grey. The mean NO₂ columns over central-eastern China (blue rectangle) are given inset in units of [10¹⁵ molecules cm⁻²], e.g., 4.8×10¹⁵ molecules cm⁻² for the top left panel. The scatterplots show the spatial correlations between the OMI retrievals and GEOS-Chem on the 0.5°×0.625° grid for central-eastern China, along with the Pearson's correlation coefficient (*r*), reduced-major-axis linear regressions, and regression slopes. Error standard deviations on the slopes were derived by the bootstrap method.

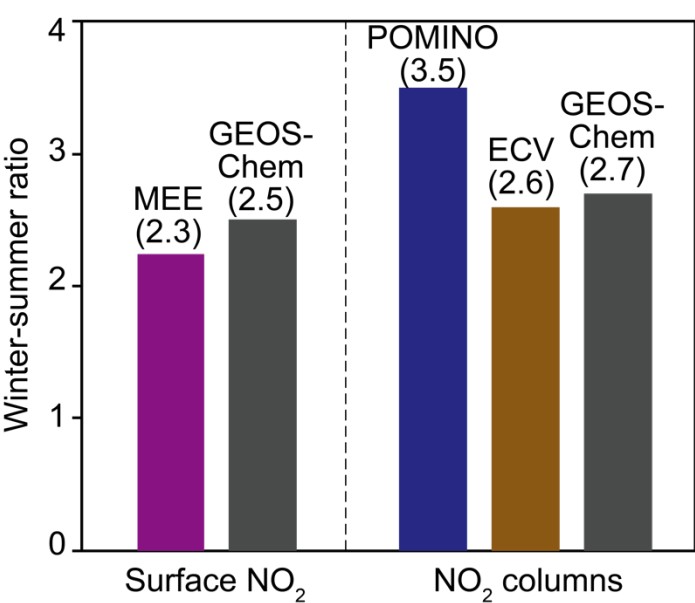

**Figure 2: Mean winter-summer ratios of NO₂ concentrations over central-eastern China.** The values are ratios of the seasonal mean NO₂ surface concentrations observed at the ensemble of MEE sites in central-eastern China, and of the mean POMINO and ECV tropospheric NO₂ columns (recalculated using the GEOS-Chem NO₂ profiles) at the MEE station locations. Central-eastern China is as defined by the rectangle in Figure 1. GEOS-Chem values sampled at the measurement locations are also shown. Observations and model results are for JJA 2017 (summer) and DJF 2016/17 (winter).

600

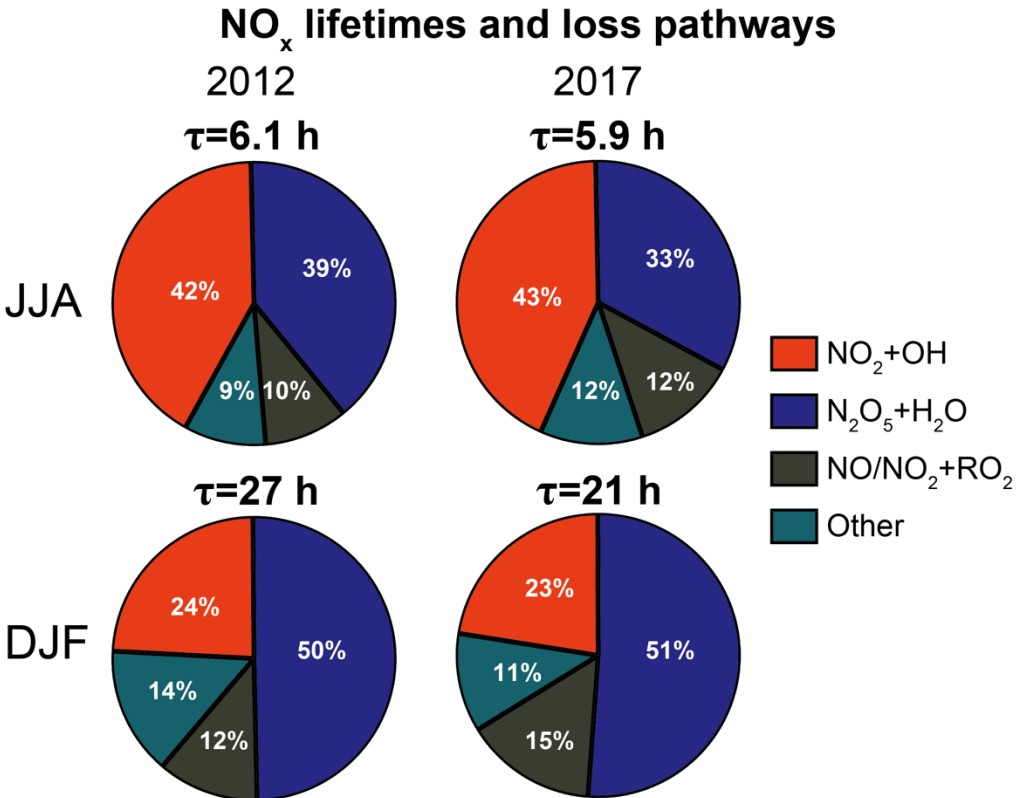

Figure 3: NO$_x$ lifetime ($\tau$) and loss pathways in the boundary layer over central-eastern China. The lifetimes are the GEOS-Chem averages for the bottom 0-2 km of atmosphere over the domain delineated in Figure 1, and the pie charts show the relative contributions of the different NO$_x$ sinks. For the lifetime calculation we define NO$_x$ as NO+NO$_2$+NO$_3$+2N$_2$O$_5$+HONO+HNO$_4$+ ClNO$_2$. Values are given for summer (JJA) and winter (DJF) of 2012 and 2017. The sink from N$_2$O$_5$+H$_2$O excludes the fraction that forms ClNO$_2$. The sink from NO/NO$_2$ + RO$_2$ is the net flux, accounting for partial recycling of the organic nitrates, and includes the contributions from peroxyacyl nitrates (PANs). The 'Other' sinks include NO$_3$+VOC reactions, NO$_2$ and NO$_3$ hydrolysis in aerosols, and NO$_x$ deposition.

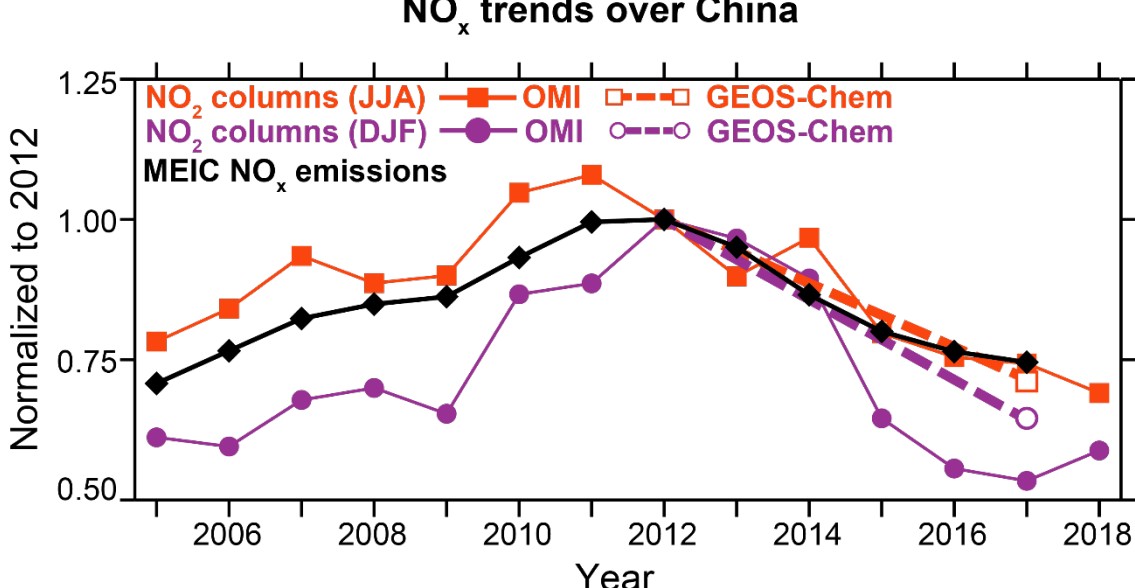

Figure 4: Trends in NO$_x$ emissions and NO$_2$ concentrations over central-eastern China. The figure shows the JJA and DJF relative trends in the OMI (ECV) tropospheric NO$_2$ columns averaged over central-eastern China (domain delineated in Figure 1) for the 2004–2018 duration of the OMI record, and the corresponding trends in annual NO$_x$ emissions estimated by the MEIC inventory. GEOS-Chem 2012–2017 trends in NO$_2$ columns are also shown. All quantities are normalized to a value of 1 in 2012.

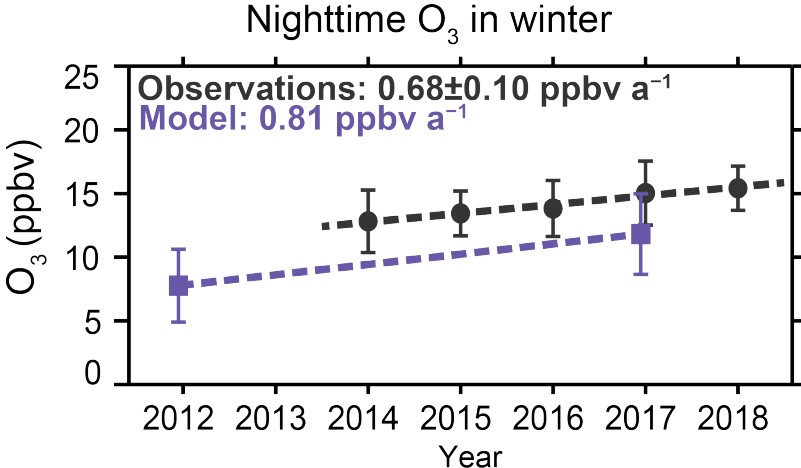

620 **Figure 5: Trend in nighttime surface ozone in China in winter. Values are DJF means over 21:00–5:00 local time for the network of sites in central-eastern China operated since 2013 by the China Ministry of Ecology and Environment (MEE). The sites are gridded on the 0.5°×0.625° GEOS-Chem grid. The symbols are averages for all grid cells containing sites and vertical bars are the standard deviation of the spatial distribution. Model values are similarly sampled and gridded over the ensemble of MEE sites operating since 2013. Trends are from an ordinary least squares regression.**