# Peer review of "Effect of changing NOx lifetime on the seasonality and long-term trends of satellite-observed tropospheric NO2 columns over China"

_Atmospheric Chemistry and Physics, 2019_

## Short Comment (SC1) · 19 Aug 2019

Dear authors, congratulations on this nice peace of work. In support of your discussion, I would like to bring your attention to a recent study on satellite-based tropospheric NO2 trends and trend reversals (1996-2017). The sharp decline over eastern China after 2011 is clearly shown on Fig. 4a and 5b. Similar results appear on more regional studies (e.g. Wang et al., 2019) and also in recent PM2.5 studies (Ma et al., 2019).

Georgoulias, A. K., van der A, R. J., Stammes, P., Boersma, K. F., and Eskes, H. J.: Trends and trend reversal detection in 2 decades of tropospheric NO2 satellite observations, Atmos. Chem. Phys., 19, 6269-6294, doi:10.5194/acp-19-6269-2019,

2019.

Ma, Z., Liu, R., Liu, Y., and Bi, J.: Effects of air pollution control policies on PM2.5 pollution improvement in China from 2005 to 2017: a satellite-based perspective, Atmos. Chem. Phys., 19, 6861-6877, doi:10.5194/acp-19-6861-2019, 2019.

Wang, C., Wang, T., and Wang, P.: The Spatial-Temporal Variation of Tropospheric NO2 over China during 2005 to 2018, Atmosphere, 10, 444, doi:10.3390/atmos10080444, 2019.

---

## Referee Comment (RC1) · Anonymous Referee #3 · 20 Aug 2019

The study examines and compares the trends of spaceborne NO2 columns and bottom-up NOx emissions (MEIC) over Eastern China. The observed NO2 trends between 2012 and 2017 appear consistent with the emission trends in summer as well as in winter, when taking into account changes in the NOx lifetimes during that period. GEOS-Chem simulations for the years 2012 and 2017 indicate very little change in the summertime NOx lifetime, due to the compensation of higher daytime OH and lower nighttime aerosol loss; in winter, however, the increasing trend in nighttime ozone appears to drive a decreasing trend of the NOx lifetime, due to the large share of the overall NOx sink being due to N2O5 hydrolysis on aerosols. These model findings appear consistent with the comparison of MEIC emissions and OMI columns.

The topic of the paper is very significant since the community and policy-makers will be interested by further information on NOx emission trends and how they relate to NO2 data. The paper is clear and well written. However, I am not entirely convinced by the robustness of the conclusions.

Major comments

For one thing, uncertainties should be better acknowledged and (wherever possible) evaluated. Given the importance of the aerosol sink in the discussion of the trends, sensitivity studies are necessary to demonstrate that the conclusion holds despite uncertainties in aerosol surface densities and especially in the N2O5 uptake coefficient. I understand that the NO2+O3 reaction might be the main limiting factor to N2O5 loss, at least during the winter, but it might not be the case during summer, and in any case it requires more than just qualitative arguments. It is worth noting that campaign data (e.g. Brown et al., JGR 114, D00F10, doi:10.1029/2008JD011679, 2009) suggested much lower N2O5 uptake coefficients than those (Bertram and Thornton) used in GEOS-Chem.

A second major comment is related to the calculation of NOx lifetimes (Figure 3), which are averages for the bottom 0-2 km of the atmosphere. I assume that the values given (about 6 hours in the summer and 21-27 hours in winter) are 24-hour averages. Two issues arise: 1) in summer, the relevant lifetime for OMI NO2 columns is the average NOx lifetime during the few (∼6) hours preceding the satellite overpass time (13:30 LT); and 2) the vertical profile of the OMI sensor sentivity should be taken into account. I know that at least 70% of the columns lies below 2 km altitude, but the sensitivity profile is very steep and very anti-correlated with the NO2 vmr profile, even below 2 km. Furthermore, the part of the column lying above 2 km altitude is non-negligible and might have very different trends from the lowermost part. I suspect that taking these effects into account will increase the relative importance of NO2+OH to the total loss, with possibly significant consequences for the trend and for the seasonal evolution.

Minor comments:

- line 112 how important is the correction using GEOS-Chem simulated concentrations of HNO3 and organic nitrates?

- why not use the ground-based NO2 measurements to evaluate the trends?

- l. 135 the NO2 hydrolysis coefficient is said to be decreased from $10^{-4}$ to $10^{-5}$. Relative to what study or what model version? Is this value used for all RH and all aerosol types? Jaeglé et al. (JGR 2018) use $10^{-4}$ and assume that the reaction makes only HONO (i.e. no HNO3), which would lead to very high HONO/NO2 ratios over China. A comment on this would be appreciated.

- what is the model performance for aerosol surface density over China?

- l. 136-138 This gives the impression that the well-known issue of the HONO missing sources has been solved, which is not the case, since many other sources have been proposed to explain the observations. What is the contribution of NO2 hydrolysis to the NOx sink in the simulations?

l. 172-174 The NO2 column measurement and the surface NO2 measurement represent different times, therefore the similar winter-summer ratios are not necessarily expected.

l. 186 Regarding the aerosol loss, a discussion of the uptake coefficient is required (see above, major comment)

l. 187-190 and Figure 4: not useful for the discussion, could possibly be dropped.

- Figure 1: I suggest to indicate on each plot, the domain-averaged NO2 column

Technical comments:

- the reference Manders et al. (2017) is missing

- the reference Bertram and Thornton (2009) is missing

- references: use journal abbreviations

---

## Author Comment (AC1) · 20 Aug 2019

Dear Aristeidis Georgoulias. Thank you for your comment and for bringing to our attention these studies on satellite-based tropospheric NO2 trends in China. We will include them in the revised manuscript.

---

## Referee Comment (RC2) · Anonymous Referee #2 · 2 Sep 2019

General Comments: Shaw et al present a combined model, satellite, and ground-based approach to disentangle the effects of changing NOx emissions and NOx lifetime on observed column NO2. The results of this study, and studies like this, are of great importance to the community seeking to utilize remote sensing approaches to infer trends in emissions. It has been well established that NOx lifetime is dependent on NOx concentration (due to the feedback on HOx) especially in the extremes of high and low [NOx] leading to strong spatial variability in NOx lifetime. There has been less focus on variability in tau(NOx) at a fixed location can impact calculated NOx emissions and the impact of NOx on tau(NOx) beyond its control on HOx.

The manuscript is well written and within the scope of ACP. I recommend that it be published following the authors attention to the following comments.

Specific Comments:

1. Model resolution: To what extent does model spatial resolution impact the results? If I am not mistaken, the model resolution is approximately 50 x 50 km in the study region. I would expect that O3 titration would display significant variability on this scale and that the mean modeled P(NO3) = k[NO2][O3], which is driving the nocturnal NOx lifetime, may not correspond to that calculated at smaller spatial scales? It would be helpful for the authors to comment on the extent to which model resolution is important and what direction the effects of resolution may have on calculations in NOx lifetime.

2. N2O5 to NO3 ratio: The N2O5 / NO3 ratio also scales with [NOx]. With decreasing NOx, this ratio decreases and L(NO3) becomes more important than L(N2O5). To what extent is this important here, or is the nocturnal NOx lifetimes essentially all limited by P(NO3) and L(N2O5 + NO3) »> P(NO3)? While this may not impact the retrieval of NOx emissions trends, it could have a sizeable effect on nitrate aerosol formation rates.

3. ClNO2 branching fraction: What is the mechanism for ClNO2 in the model? What is the distribution of ClNO2 branching fractions? Does this change in time? If 30-50% of NOx is lost to N2O5, ClNO2 has the potential to return half of this. A short section on the parameterization used and the uncertainty in this (most measurements show that parameterizations of ClNO2 branching fractions are much larger than observations) should be included. I appreciate that NOx lifetime may not be that dependent on aerosol surface area, but the net NOx removal is certainly dependent on the ClNO2 branching fraction.

---

## Author Response (AR1)

**We thank the referees for their insightful comments. Our point-by-point responses to the referee comments and the revisions to the manuscript are as follows:**

Response to Referee # 2
General Comments: Shah et al present a combined model, satellite, and ground-based approach to disentangle the effects of changing NOx emissions and NOx lifetime on observed column NO2. The results of this study, and studies like this, are of great importance to the community seeking to utilize remote sensing approaches to infer trends in emissions. It has been well established that NOx lifetime is dependent on NOx concentration (due to the feedback on HOx) especially in the extremes of high and low [NOx] leading to strong spatial variability in NOx lifetime. There has been less focus on variability in tau(NOx) at a fixed location can impact calculated NOx emissions and the impact of NOx on tau(NOx) beyond its control on HOx. The manuscript is well written and within the scope of ACP. I recommend that it be published following the authors attention to the following comments.

Specific Comments:
1. Model resolution: To what extent does model spatial resolution impact the results? If I am not mistaken, the model resolution is approximately 50 x 50 km in the study region. I would expect that O3 titration would display significant variability on this scale and that the mean modeled P(NO3) = k[NO2][O3], which is driving the nocturnal NOx lifetime, may not correspond to that calculated at smaller spatial scales? It would be helpful for the authors to comment on the extent to which model resolution is important and what direction the effects of resolution may have on calculations in NOx lifetime.
Following the referee's suggestion, we have added a discussion about this in the manuscript.
Added text:
Line 210 - Sub-grid scale variability in $NO_x$ and oxidant concentrations can also affect the modeled $NO_x$ lifetimes because of the non-linearity of $NO_x$ chemistry. For example, $NO_x$ lifetime within a concentrated $NO_x$ plume at night can be long because of oxidant depletion, and assuming that the plume is instantaneously diluted within the model grid cell can result in a shorter $NO_x$ lifetime (Brown et al., 2012). However, studies that explicitly treat sub-grid scale plumes suggest that this effect can be important locally near large sources but is small at the regional scale (Karamchandani et al., 2011).

2. N2O5 to NO3 ratio: The N2O5 / NO3 ratio also scales with [NOx]. With decreasing NOx, this ratio decreases and L(NO3) becomes more important than L(N2O5). To what extent is this important here, or is the nocturnal NOx lifetimes essentially all limited by P(NO3) and L(N2O5 + NO3) >> P(NO3)? While this may not impact the retrieval of NOx emissions trends, it could have a sizeable effect on nitrate aerosol formation rates.
We do see an increase in $NO_x$ loss via the $NO_3$+VOC reactions at lower $NO_x$ emissions. However, the contribution of $L(NO_3)$ to the total $NO_x$ loss is less than 5% (higher in summer than in winter) because of high $NO_2$ concentrations and the effect of the change in $L(NO_3)$ on the overall $NO_x$ lifetime is small. We have added the following text to the manuscript:

Added text:
Line 246 - At lower $NO_x$ emissions, $NO_x$ loss by $NO_3$+VOC reactions also increased relative to $NO_x$ loss by $N_2O_5$ hydrolysis, because of an increase in the ratio of $NO_3$ to $N_2O_5$ concentrations. However, the contribution of $NO_3$+VOC reactions to the total $NO_x$ loss remains less than 5% because of high $NO_2$.

3. ClNO2 branching fraction: What is the mechanism for ClNO2 in the model? What is the distribution of ClNO2 branching fractions? Does this change in time? If 30-50% of NOx is lost to N2O5, ClNO2 has the potential to return half of this. A short section on the parameterization used and the uncertainty in this (most measurements show that parameterizations of ClNO2 branching fractions are much larger than observations) should be included. I appreciate that NOx lifetime may not be that dependent on aerosol surface area, but the net NOx removal is certainly dependent on the ClNO2 branching fraction.
In the model, $ClNO_2$ forms by $N_2O_5$ hydrolysis only on sea-salt aerosols, with a fixed branching ratio of 1. When we calculate the $NO_x$ lifetime, we include $ClNO_2$ in the $NO_x$ burden and do not consider $N_2O_5$+$H_2O$/$Cl^-$ → $ClNO_2$ as a $NO_x$ loss process. We have clarified these two points in the revised manuscript and added a discussion on the effect of this model assumption on the $NO_x$ lifetime.

Added text:
Line 137 - $N_2O_5$ hydrolysis produces $HNO_3$ and $ClNO_2$ on sea-salt aerosols with a 1:1 branching ratio (reaction R8) and only $HNO_3$ on other aerosol types (reaction R7).

Line 205 - We assume that the yield of $ClNO_2$ from $N_2O_5$ hydrolysis for all aerosols other than sea-salt is 0, which is lower than the average value of ~0.3 estimated by field studies in China (McDuffie et al., 2018a; Tham et al., 2018). If we had assumed a higher $ClNO_2$ yield in the model, the loss of $NO_x$ by $N_2O_5$ hydrolysis would be slower, particularly in winter when it is limited by ozone and the $HNO_3$ and $ClNO_2$ branches compete for the limited $N_2O_5$, and the daytime $NO_x$ loss would increase because of the additional daytime $NO_2$ from $ClNO_2$ photolysis.

Caption of Fig. 3: The sink from $N_2O_5$+$H_2O$ excludes the fraction that forms $ClNO_2$.

Response to Referee # 3

The study examines and compares the trends of spaceborne NO2 columns and bottom-up NOx emissions (MEIC) over Eastern China. The observed NO2 trends between 2012 and 2017 appear consistent with the emission trends in summer as well as in winter, when taking into account changes in the NOx lifetimes during that period. GEOS-Chem simulations for the years 2012 and 2017 indicate very little change in the summertime NOx lifetime, due to the compensation of higher daytime OH and lower nighttime aerosol loss; in winter, however, the increasing trend in nighttime ozone appears to drive a decreasing trend of the NOx lifetime, due to the large share of the overall NOx sink being due to N2O5 hydrolysis on aerosols. These model findings appear consistent with the comparison of MEIC emissions and OMI columns.

The topic of the paper is very significant since the community and policy-makers will be interested by further information on NOx emission trends and how they relate to NO2 data. The paper is clear and well written. However, I am not entirely convinced by the robustness of the conclusions.

Major comments

For one thing, uncertainties should be better acknowledged and (wherever possible) evaluated. Given the importance of the aerosol sink in the discussion of the trends, sensitivity studies are necessary to demonstrate that the conclusion holds despite uncertainties in aerosol surface densities and especially in the N2O5 uptake coefficient. I understand that the NO2+O3 reaction might be the main limiting factor to N2O5 loss, at least during the winter, but it might not be the case during summer, and in any case it requires more than just qualitative arguments. It is worth noting that campaign data (e.g. Brown et al., JGR 114, D00F10, doi:10.1029/2008JD011679, 2009) suggested much lower N2O5 uptake coefficients than those (Bertram and Thornton) used in GEOS-Chem.

Following the referee's suggestion, we now include a discussion of the uncertainties and the results of a new sensitivity simulation that we conducted to evaluate the effects of the main uncertainties. The following text was added to the modified manuscript:

Added text:

Line 200 - The modeled $NO_x$ lifetimes can be affected by uncertainties in the modeled aerosol surface area, $\gamma_{N2O5}$, and the yield of $ClNO_2$. We find that the GEOS-Chem aerosol concentrations at the surface are about 30% higher than observations from the MEE network. On the other hand, GEOS-Chem's $\gamma_{N2O5}$ values are 25% lower on average than the observation-based estimates from the WINTER campaign (Jaeglé et al., 2018; McDuffie et al., 2018b). We tested the sensitivity of the modeled $NO_x$ lifetime to the aerosol surface area and $\gamma_{N2O5}$ with a separate simulation and find that a 30% change in either quantity changes the $NO_x$ loss by $N_2O_5$ hydrolysis by only 8% in summer and less than 2% in winter. We assume that the yield of $ClNO_2$ from $N_2O_5$ hydrolysis for all aerosols other than sea-salt is 0, which is lower than the average value of ~0.3 estimated by field studies in China (McDuffie et al., 2018a; Tham et al., 2018). If we had assumed a higher $ClNO_2$ yield in the model, the loss of $NO_x$ by $N_2O_5$ hydrolysis would be slower, particularly in winter when it is limited by ozone and the $HNO_3$ and $ClNO_2$ branches compete for the limited $N_2O_5$, and the daytime $NO_x$ loss would increase because of the additional daytime $NO_2$ from $ClNO_2$ photolysis. Sub-grid scale variability in $NO_x$ and oxidant concentrations can also affect the modeled $NO_x$ lifetimes because of the non-linearity of $NO_x$ chemistry. For example, $NO_x$

lifetime within a concentrated $NO_x$ plume at night can be long because of oxidant depletion, and assuming that the plume is instantaneously diluted within the model grid cell can result in a shorter $NO_x$ lifetime (Brown et al., 2012). However, studies that explicitly treat sub-grid scale plumes suggest that this effect can be important locally near large sources but is small at the regional scale (Karamchandani et al., 2011).

A second major comment is related to the calculation of NOx lifetimes (Figure 3), which are averages for the bottom 0-2 km of the atmosphere. I assume that the values given (about 6 hours in the summer and 21-27 hours in winter) are 24-hour averages. Two issues arise: 1) in summer, the relevant lifetime for OMI NO2 columns is the average NOx lifetime during the few (~6) hours preceding the satellite overpass time (13:30 LT); and 2) the vertical profile of the OMI sensor sensitivity should be taken into account. I know that at least 70% of the columns lies below 2 km altitude, but the sensitivity profile is very steep and very anti-correlated with the NO2 vmr profile, even below 2 km. Furthermore, the part of the column lying above 2 km altitude is non-negligible and might have very different trends from the lowermost part. I suspect that taking these effects into account will increase the relative importance of NO2+OH to the total loss, with possibly significant consequences for the trend and for the seasonal evolution.

The seasonal NOx lifetimes are 24-hour mean values. We have clarified this in text manuscript. We also agree that because of the shorter summer lifetime the daytime loss processes affect OMI $NO_2$ columns much more than those at night. We have now stated this this in the revised manuscript.
To test how sensitive the lifetime is to the column height, we recalculated the $NO_x$ lifetime for the full tropospheric $NO_2$ column and took into account the vertical sensitivity of the retrieval. We find minor differences compared to the 0–2 km mean $NO_x$ lifetimes and loss pathways ($NO_2$+OH does increase a little), because bulk of the $NO_x$ column is located below 2 km. We have included the figure showing the tropospheric column $NO_x$ lifetime and loss pathways in the Supplement and have referenced it in the main text.

Added text (underlined):
Line 186 - We find in GEOS-Chem that the seasonal variation in $NO_2$ columns is mainly driven by the $NO_x$ lifetime in the boundary layer, which increases from 5.9 h in summer to 21 h in winter (24-h mean lifetimes for 2017, Fig. 3). We find little difference between the boundary layer and the tropospheric column $NO_x$ lifetimes (Supplement Fig. S2) since most of the column is in the boundary layer. In summer, $NO_x$ is lost mostly through oxidation by OH in daytime (43%) and through $N_2O_5$ hydrolysis forming $HNO_3$ at night (33%). The daytime oxidation processes have a larger effect on the afternoon $NO_2$ columns because of the short $NO_x$ lifetime in summer.

Added supplemental figure:

[Figure]

Figure S2: $NO_x$ lifetime ($\tau$) and loss pathways for the tropospheric column over central-eastern China. The lifetimes are the GEOS-Chem averages for the tropospheric column weighted by the OMI averaging kernel over the domain delineated in Figure 1, and the pie charts show the relative contributions of the different $NO_x$ sinks. For the lifetime calculation we define $NO_x$ as $NO+NO_2+NO_3+2N_2O_5+HONO+HNO_4+ ClNO_2$. Values are given for summer (JJA) and winter (DJF) of 2012 and 2017. The sink from $NO/NO_2 + RO_2$ is the net flux, accounting for partial recycling of the organic nitrates, and includes the contributions from peroxyacyl nitrates (PANs). The 'Other' sinks include $NO_3$ + VOC reactions, $NO_2$ and $NO_3$ hydrolysis in aerosols, and $NO_x$ deposition.

Minor comments:
- line 112 how important is the correction using GEOS-Chem simulated concentrations of HNO3 and organic nitrates?
The correction for $HNO_3$ and organic nitrates decreases the seasonal-mean $NO_2$ concentrations over eastern China by 25% in summer and 6% in winter. We have specified this in the revised manuscript.

Added text:
Line 116 - The correction for $HNO_3$ and organic nitrates decreases the seasonal-mean $NO_2$ concentrations over eastern China by 25% in summer and 6% in winter.

- why not use the ground-based NO2 measurements to evaluate the trends?
We did not use the ground-based $NO_2$ measurements to evaluate the trends because the stations are located in dense urban areas and are more sensitive to trends of vehicular $NO_x$ emissions than trends in NOx emissions from industries and power plants. Indeed, Zheng et al. (2018) showed that trends the surface $NO_2$ concentrations were flatter than trends in OMI $NO_2$ columns. We have cited this result from Zheng et al. (2018) in the revised text.

Added text:

Line 224 - Zheng et al. (2018) also showed consistency in the trends of OMI $NO_2$ columns and the MEIC $NO_x$ emissions, but found that the $NO_2$ trends at the MEE surface sites are flatter because the sites are urban and more sensitive to vehicular $NO_x$ emissions.

- l. 135 the NO2 hydrolysis coefficient is said to be decreased from $10^{-4}$ to $10^{-5}$. Relative to what study or what model version? Is this value used for all RH and all aerosol types? Jaeglé et al. (JGR 2018) use $10^{-4}$ and assume that the reaction makes only HONO (i.e. no HNO3), which would lead to very high HONO/NO2 ratios over China. A comment on this would be appreciated.

We have added the reference for the original $\gamma$ values, and have added a comment regarding the difference of the $\gamma$ and products of $NO_2$ hydrolysis with those in Jaeglé et al. (2018). Please see the modified text below.

- l.136-138 This gives the impression that the well-known issue of the HONO missing sources has been solved, which is not the case, since many other sources have been proposed to explain the observations.

We agree with the reviewer that the issue of missing HONO sources is an open question. We have revised the text to reflect this better. Please see the modified text below.

What is the contribution of NO2 hydrolysis to the NOx sink in the simulations?

$NO_2$ hydrolysis is a minor sink of $NO_x$, contributing between 1% (2011 summer) and 7% (2011 winter).

Added text (underlined):

Line 138 - Uniform values of $\gamma_{NO_3}$ and $\gamma_{NO_2}$ are used for all aerosol types and all RH conditions. $\gamma_{NO_3}$ is taken to be $1\times10^{-3}$ following Jacob (2000). $\gamma_{NO_2}$ for the hydrolysis reaction (R11) is decreased from $1\times10^{-4}$ (Jacob, 2000) to $1\times10^{-5}$ on the basis of laboratory measurements (Bröske et al., 2003; Stemmler et al., 2007; Tan et al., 2016). This decrease of $\gamma_{NO_2}$ yields a 24 h mean wintertime HONO/$NO_2$ molar concentration ratio of 0.035 over eastern China in GEOS-Chem, consistent with the observed range of 0.015–0.071 (Hendrick et al., 2014; Spataro et al., 2013; Wang et al., 2017, 2013). The GEOS-Chem HONO/$NO_2$ ratio is likely low because of unknown sources of HONO particularly during the day (Kleffmann, 2007; Spataro and Ianniello, 2014). For the WINTER campaign, Jaeglé et al. (2018) used a $\gamma_{NO_2}$ of $1\times10^{-4}$ but assumed that the reaction produces only HONO. Using this $\gamma_{NO_2}$ for eastern China would lead to a significant overestimate of the observed HONO/$NO_2$ ratio.

- what is the model performance for aerosol surface density over China?
GEOS-Chem overestimated aerosol surface area by 30% compared to ground-level measurements at the MEE sites. Specified this on Line 201.

- l.172-174 The NO2 column measurement and the surface NO2 measurement represent different times, therefore the similar winter-summer ratios are not necessarily expected.
We agree that the winter-summer ratios of $NO_2$ columns and surface $NO_2$ need not be similar as the measurement times differ. We have modified the text to reflect that the similarity exists despite the different measurement times.

Revised text:

Line 180 -

GEOS-Chem shows similar ratios for the afternoon NO$_2$ columns and 24-h mean surface NO$_2$, despite different averaging times.

- l.186 Regarding the aerosol loss, a discussion of the uptake coefficient is required (see above, major comment)
We have added a discussion of the uptake coefficient in the revised manuscript as described in the response to major comment #1 (above).

- l.187-190 and Figure 4: not useful for the discussion, could possibly be dropped.
We have moved Fig. 4 to the Supplement, following the referee's suggestion. We have retained the text on lines 187-190, because it clarifies why the NO$_x$ lifetime derived from the decay of NO$_2$ concentrations away from a source cannot be used to verify the simulated NO$_x$ chemical lifetime for our study region.

- Figure 1: I suggest to indicate on each plot, the domain-averaged NO2 column
Following the referee's suggestion, we have added the domain-averaged NO$_2$ column in the figure.

Technical comments:
- the reference Manders et al. (2017) is missing
The Manders et al. (2017) reference is on Line 461.
- the reference Bertram and Thornton (2009) is missing
We have now added the reference for Bertram & Thornton (2009)
- references: use journal abbreviations
We have edited the References and use journal abbreviations.

[revised manuscript text omitted]